# The Role of Walking in the Relationship between Catastrophizing and Fatigue in Women with Fibromyalgia

**DOI:** 10.3390/ijerph19074198

**Published:** 2022-04-01

**Authors:** Lucía Sanromán, Patricia Catalá, Carmen Écija, Carlos Suso-Ribera, Jesús San Román, Cecilia Peñacoba

**Affiliations:** 1Department of Psychology, Rey Juan Carlos University, 28922 Alcorcón, Spain; l.sanromanc@alumnos.urjc.es (L.S.); patricia.catala@urjc.es (P.C.); carmen.ecija@urjc.es (C.É.); 2Department of Basic and Clinical Psychology and Psychobiology, Jaume I University, 12006 Castellón de la Plana, Spain; susor@uji.es; 3Department of Medical Specialties and Public Health, Rey Juan Carlos University, 28922 Alcorcón, Spain; jesus.sanroman@urjc.es

**Keywords:** chronic pain, physical activity, physical symptoms, cognitive process, observational descriptive study

## Abstract

Walking is one of the most beneficial treatments for fibromyalgia patients. However, adherence to walking behavior is low due to the initially associated symptoms (including pain and fatigue). Although the association of catastrophism with greater symptoms is known, the results regarding fatigue have not always been consistent. Nevertheless, it is unknown whether the association between catastrophism and fatigue could, in turn, be conditioned by whether the patients walk or not. Therefore, our goal was to explore the moderating effect of walking on the association between catastrophizing and fatigue in patients with fibromyalgia. A cross-sectional study was carried out with 203 women with fibromyalgia. We used the Multidimensional Fatigue Inventory to assess fatigue and the Pain Catastrophizing Scale to assess pain catastrophizing (differentiating between its three dimensions). An ad hoc item was used to evaluate walking (moderator). Lower scores for fatigue and pain catastrophizing were found among patients who walked versus those who did not. Walking moderated the relationship between rumination and fatigue (Beta = 0.16, t = 1.96, *p* = 0.049) and between magnification and fatigue (Beta = 0.22, t = 21.83, *p* = 0.047). Helplessness showed no direct or interaction effect for fatigue. Nevertheless, higher rumination and magnification were associated with higher fatigue only in patients who walked. Therefore, to promote adherence to walking and reduce the effects of catastrophizing on fatigue, it seems necessary to manage rumination and magnification among patients who walk.

## 1. Introduction

Fibromyalgia is a disorder characterized by chronic widespread pain and frequent comorbid symptoms, such as sleep disturbances, cognitive impairment, emotional disorders and fatigue. It was not until 2010, however, that the diagnostic criteria were changed and the clinical significance of these comorbid symptoms were clearly recognized [1]. With these new criteria, fatigue, unrefreshing sleep, cognitive problems and a group of somatic symptoms are also added. Therefore, the change in the criteria meant that pain ceased to be the only diagnostic symptom or necessarily the most relevant. Another important modification of the criteria meant that the way pain was assessed changed from trigger points to areas of pain. Because of how the previous diagnostic criteria were designed, for years, pain was the most widely investigated symptom in fibromyalgia. However, another symptom, fatigue, has often been reported by patients as one of the most disturbing and disabling symptoms [2,3].

Fatigue is a highly prevalent and persistent symptom among fibromyalgia patients [4], and it contributes significantly to functional limitations among sufferers [5]. From a rehabilitation and prevention point of view, one of the main treatment aims in patients with fibromyalgia is to maintain physical function and avoid the disabilities that can arise from the disorder [6,7]. In this sense, fibromyalgia treatments that have included activities and physical exercise as one of their therapeutic aims have shown positive effects on patients’ health outcomes [8,9,10], including a significant reduction in fatigue [11,12]. Walking, which is a form of physical exercise, has significant advantages over other physical practices due to its easy implementation and accessibility, its practice is associated with reduced fatigue, pain and functional limitations, making walking an excellent treatment option for people with fibromyalgia [13,14,15,16].

In the context of physical exercise and within models of fear of movement [17], pain catastrophizing has been repeatedly associated with more severe symptomatology and a poorer adaptation to fibromyalgia [18,19]. Pain catastrophizing is a very prevalent cognitive process among fibromyalgia patients [20,21] characterized by a pessimistic and exaggerated interpretation of current or anticipated pain sensations [22] which leads to the pain being perceived worse than it truly is.

Pain catastrophizing is composed of three elements: magnification is related to the disproportionate perception of pain situations and expectations of negative results. Rumination includes ruminating thoughts, worry and an inability to inhibit thoughts related to pain; and helplessness refers to the ability to control it [23]. This biased interpretation of events has been argued to lead to avoidance of movement and, ultimately, to poor overall physical status [17]. Initial studies regarding pain catastrophizing were focused solely on its association to pain; however, recent studies have highlighted its relation to other symptoms of fibromyalgia, such as functionality and disability [7,24,25]. The specific relationship between pain catastrophizing and fatigue has been scarcely studied in the literature and is not well established in fibromyalgia [26,27,28,29,30]. While some research has shown that both variables are related [29], finding catastrophizing to be a predictor of general fatigue [27,28] capable of producing differential effects on this symptom, other more recent literature suggests there is no relationship between catastrophizing and fatigue [30].

Past research has shown that the relationship between pain catastrophizing and physical health outcomes in people with chronic musculoskeletal pain might be contextually determined by patient characteristics, for example, by pain severity [31]. This is the theoretical basis that underlies the models of psychological flexibility [32] that have shown that the relationships between psychological variables and function are not linear per se and may be affected by additional variables that give rise to complex models in chronic pain [33,34,35]. Given the known benefits of physical exercise in fibromyalgia patients [36,37,38] and of walking in particular [39,40], and the low adherence to this guideline largely due to the symptoms experienced [13], it would be of interest to analyze the extent to which pain catastrophism can have a differential role, specifically on fatigue (a symptom barely studied), in patients who walk (in comparison to those who do not). Specifically, our goal has been to test whether adherence to walking is an important contextual factor to be considered in the relationship between pain catastrophizing and fatigue. Based on previous studies [28,41], it would be expected that, in the worst-case scenario, that is to say when the contextual variables are especially limiting (i.e., when patients do not walk at all), individual status (i.e., fatigue) will be so unfavorable that there will be little room for pain catastrophizing to contribute to it. In contrast, when patients adhere to walking, we expect fatigue to be high if they walk while using biased interpretation (i.e., catastrophizing).

The findings might have important repercussions on the design of programs aimed at maintaining walking as a form of physical exercise due to its known benefits [38,39], by trying to reduce the impact of catastrophizing on fatigue, since this symptom is highly disabling [3,4] but has rarely been the focus of research [2].

## 2. Materials and Methods

### 2.1. Participants

In order to participate, all participants had to have a diagnosis of fibromyalgia, according to the diagnostic criteria for fibromyalgia of the American College of Rheumatology [1,42]. In addition, to meet our inclusion criteria participants also had to be female (for homogeneity purposes because most fibromyalgia patients are females), be over 18 years of age, have the physical and mental ability to provide informed consent and to complete the surveys, and have a medical recommendation to walk and/or not present physical comorbidity that prevented walking.

The sample, initially, included 234 women with fibromyalgia who were between the ages of 30 and 78, with a mean of 57.03 and a standard deviation (SD) of 9.07. Most participants had completed only primary (53.2%) or secondary studies (26.4%). The majority of them were married or in a stable relationship at the time of assessment (53%). The remaining participants were either single (11%) or divorced/widowed (36%). Only a small percentage of the participants were working when the study was conducted (12%). The rest of them were homemakers (33.8%), retired (32%), on sick leave (10%) or unemployed (12.1%). The patients had experienced fibromyalgia for an average of 12.14 years (SD = 8.45; 1–46 years range). Their average pain severity score was 7.15 (SD = 1.52) on a scale ranging from 0 to 10, with higher scores indicating more severe pain.

### 2.2. Instruments

#### 2.2.1. Study Variables

General fatigue. The Spanish version of the Multidimensional Fatigue Inventory (MFI) [43] was used to assess fatigue. This is a questionnaire with 20 items scored on a five-point Likert response scale with anchors 1 = “Completely agree” to 5 = “Strongly disagree”. The general fatigue dimension, which is composed of 4 items and has a score range of 4 to 20, was selected for the present study. The other dimensions in the questionnaire are physical fatigue, mental fatigue, reduced activity and reduced motivation. Higher scores indicate greater presence of fatigue. The internal consistency of the general fatigue scale in the present study was 0.76.

Pain Catastrophizing. The Spanish version of the Pain Catastrophizing Scale (PCS) [23] has 13 items grouped into 3 dimensions: rumination, magnification and helplessness. Responses are scored using a 5-point Likert scale with anchors 0 = “Not at all” to 4 = “All the time”. Higher scores indicate a higher tendency to catastrophize pain. The Cronbach’s alpha values in the present study were 0.78 for magnification, 0.87 for rumination, 0.89 for helplessness and 0.94 for total pain catastrophizing.

Walking. The walking behavior proposed by Gusi et al. [44] for fibromyalgia patients was selected (“at least 60 min in bouts of 20 min, with a small rest between bouts, four times a week, over a minimum of six consecutive weeks”). However, the minimum daily time was reduced to 30 min and at least 2 days a week because the targeted population was highly sedentary and because of the difficulties in adherence to physical exercise in patients with fibromyalgia [39,45]. An ad hoc item was used to assess whether they adhered to walking according to the prescribed pattern (1 = “yes”/0 = “no”).

#### 2.2.2. Covariables

Pain: Four items assessing maximum, minimum and the average pain intensity during the last 7 days and pain at that moment in a numerical rating scale were used (0 = no pain, and 10 = the worst pain you can imagine) [46,47]. The internal consistency in the present study was 0.86.

Disability: To assess disability, the total score of the Spanish adaptation of the Fibromyalgia Impact Questionnaire-Revised (FIQ-R) was used [48]. Items in the FIQ-R are answered on an 11-point numerical scale ranging from 0 to 10, where higher scores represent higher disability. Total FIQ-R scores range from 0 to 100. Spanish FIQ-R has shown high internal consistency (Cronbach’s α above 0.91) [48]. The Cronbach’s alpha value in the present study was 0.89.

### 2.3. Procedure

This study is part of a larger project (PI17/00858) that mainly aims, among patients with fibromyalgia, to identify the role of catastrophism, fear of pain and perception of self-efficacy in the preference and setting of goals related to exercise and walking behavior. This manuscript is part of the first phase of this study. We collected a convenience sample by contacting several patients’ associations of fibromyalgia in various Spanish regions. These associations require members to have been diagnosed with fibromyalgia, although all diagnoses were independently confirmed by the research team [1,42]. In total, 268 participants agreed to participate in the study and met the initial inclusion criteria. Finally, effective responses were obtained from 234 patients (25 patients did not attend the scheduled assessment appointment, 6 questionnaires were left blank, and 3 questionnaires contained a large amount of missing data that could not be retrieved because the participants could no longer be reached). The study followed the ethical principles for research with human participants (Helsinki declaration) and was approved by the (blinded for review) University Ethics Committee. The patients were contacted by phone to arrange an appointment to complete the questionnaires in a personalized and face-to-face manner in a single session. The assessment sessions were carried out in rooms equipped for this purpose with adequate environmental conditions of light and temperature.

The assessment sessions lasted approximately 45 min. First, a structured interview was carried out to assess demographic and clinical variables (i.e., time since diagnosis, age, work status, educational level, etc), pain intensity [46], medical recommendation to walk and/or absence of comorbidity that would impair the ability to walk and drug prescriptions (participants were asked to bring their medical history). Furthermore, they were also asked about whether they walked for physical exercise as per the guidelines set by Gusi et al. [44]. After the interview, the participants were given the self-report questionnaires regarding disability (FIQ-R) [48], catastrophism (PCS) [23] and fatigue (MFI) [43]. The researchers (psychologists) read the instructions and the participants filled out the questionnaires. Moreover, to avoid biases caused by fatigue in the filling out of the questionnaires, the order in which they were presented was altered among the participants. The patients did not receive financial compensation for their participation.

### 2.4. Statistical Analysis

The Statistical Package for Social Sciences IBM SPSS for Windows, version 22.0 (Armonk, NY, USA) [49] was used for analyses.

Before performing the analyses, we analyzed the possible outliers through the stem plot. Four cases were excluded since they presented mean scores of less than 8 in general fatigue (dependent variable). The remaining scores were between 9 and 20.

Based on our objectives, the main analyses were of the catastrophism and fatigue variables, taking walking as a moderator. To control the effects of the covariables, previous bivariate analyses were carried out in relation to pain, disability and medication used to manage pain, so as to assess their inclusion in the moderation analyses.

First, descriptive and bivariate Pearson correlation analyses were performed. Then, an analysis of mean differences in catastrophizing (and its dimensions), fatigue, pain intensity and disability was conducted based on adherence to walking. The effect of prescription drugs on pain management was also assessed (opioids and non-opioid painkillers and anti-inflammatory medication), both in relation to the continuous variables (catastrophizing, fatigue, pain intensity and disability) using Student’s *t* tests, and in relation to walking (chi-square test). Effect sizes were calculated.

Finally, the moderation analyses were conducted with model 1 from the PROCESS Macro version 3.4 (New York, NY, USA, Andrew F. Hayes) [50]. Walking (yes/no) was used as the moderator, catastrophizing as an independent variable, and fatigue as the outcome. The role of the covariables (i.e., disability) was also taken into account in the analyses. Pain catastrophizing was centered before the analyses to make the regression coefficients clearer. Centering consists of subtracting the mean from the predictors to rescale them. Specifically, by doing this the regression coefficients will reveal the effect when the remaining predictors have a value of ‘0’ [51]. This is useful in terms of interpretation, but of course creates ‘artificial’ scores by rescaling the predictors. Centering has no effect on model fit (R2), significance tests and standardized slope values.

Four models were tested, one for each catastrophizing dimension (total score and three dimensions). Statistical significance was set at an alpha level of 0.05. In the moderation analyses, we used values in conditional tables, which are the 16th, 50th and 84th percentiles. When a moderation was found to be significant, these cut-offs were used to calculate conditional effects (i.e., effects of an independent variable on an outcome for different values of a moderator). In the post hoc analyses, non-centered variables were used to facilitate the interpretation of results. Uncentered variables reflect the ‘real’ scores with which readers will be familiar.

## 3. Results

### 3.1. Means, Standard Deviations and Pearson Correlations between Continuous Study Variables and Covariables

Table 1 shows the means, SDs and Pearson correlations between the study variables. Statistically significant correlations were found between all variables. The strongest correlations (>0.70) were found between the different dimensions of pain catastrophism. The weakest correlations were revealed between fatigue and pain catastrophism (total score and the three dimensions).

Regarding the possible covariables, statistically significant correlations were found between pain, disability, fatigue, catastrophizing and its dimensions. The strongest correlations were found between disability and the study variables.

### 3.2. Mean Differences in Study Variables and Covariables as a Function of Adherence to Walking

More than half of the patients (58.15%) adhered to walking behavior as a way of exercising. Fatigue, rumination, helplessness and pain catastrophism were higher in patients who did not adhere to walking, as shown in Table 2. The largest effect sizes were found for helplessness and general fatigue (both with moderate effect sizes).

In relation to the covariables, disability and pain were higher in patients who did not adhere to walking exercise. The largest effect size was found for disability (moderate effect size).

### 3.3. Mean Differences in Study Variables and Covariables as a Function of Prescription Painkillers

Fatigue, rumination, magnification, helplessness and pain catastrophism were higher in patients who took painkillers, as shown in Table 3. The largest effect sizes were found for rumination.

Regarding the covariables, disability and pain were higher in patients who took painkillers. The largest effect size was found for disability. Walking was not related to painkillers (x^2^ = 1.135 *p* = 0.29).

### 3.4. Multivariate Linear Regression and Moderation Analyses

The previous bivariate analyses showed the need to control both the intensity of the pain and disability in the moderation analyses. In relation to these variables, given the higher correlation found between disability in relation to the main outcome variables, it was decided to include disability as a covariable in the moderation analysis. This decision was also conceptually backed up, as disability, assessed using the FIQ-R [48], contains different scales, including symptoms, among which is pain. Regarding medication, the frequent use of painkillers was prevalent in the majority of the sample (88%). Because of this, to control for this variable, it was decided to carry out the moderation analysis only among the sample that has been prescribed this type of medication (*n* = 203).

Table 4 shows the results of the regression analyses, including walking as a moderator. The prediction of fatigue from catastrophizing, walking and their interaction evidenced significant direct contributions of walking (in all cases, except for helplessness). Regarding the moderation analyses, the results revealed that walking moderated the relationship between rumination and fatigue (Beta = 0.16, t = 1.96, *p* = 0.49), as well as the association between magnification and fatigue (Beta = 0.22, t = 1.83, *p* = 0.047). The assessed models predicted significant variance of fatigue (all *p* < 0.001), which ranged from 13% to 17%.

In relation to the influence of disability as a covariable, the results showed a direct and positive contribution in all the interaction models (*p* < 0.001) of fatigue.

Post hoc analyses were used to analyze significant moderations in greater depth and are presented in Table 5. These analyses revealed that rumination and magnification contributed to increased fatigue only when patients engaged in walking behavior.

## 4. Discussion

This study aimed to provide further evidence for the relationship between pain catastrophism and fatigue in patients with fibromyalgia by exploring the contextual (i.e., moderating) role of adherence to walking in this relationship. Past research has shown that low adherence to exercise, such as walking, imposes a significant physical burden on patients with fibromyalgia, including an increased sense of overall fatigue [13,14,15,16]. Past research has also evidenced that the contribution of psychological variables, such as pain catastrophizing, to physical outcomes might decrease when the patient’s status is more severe, for example, for those with greater pain severity [31]. Our results provide an interesting novel insight into this complex and contextually determined relationship between psychological factors and outcomes. In general, the results found support our hypothesis, specifically in relation to rumination and magnification. Both dimensions contributed to fatigue only when the context was favorable (i.e., high adherence to walking). The results are discussed in the context of the psychological flexibility model and personalized interventions. Furthermore, the fact that these results were not found in relation to helplessness shows how complex catastrophism truly is [52] and the possibility that the different components involved do not have the same role [25,52,53,54] as has been suggested in recent studies [24]. Therefore, it is important that future studies focus on catastrophism take into account specifically all its components, not in a general manner, as has been done in most studies.

To date, pain has been the most studied outcome of catastrophism, and fatigue has been studied less frequently. Additionally, it has been assumed that the relationship between pain catastrophizing and outcomes (i.e., fatigue) is linear, that is to say, it is not influenced by other personal, behavioral or contextual variables. The results of the present study show that while a modest bivariate association exists between pain catastrophism and fatigue, this association is no longer significant when adherence to walking and its interactions with catastrophizing are considered. This suggests that the relationship between rumination, magnification and fatigue is largely explained by adherence to walking, which was confirmed in the post hoc analyses of conditional effects after the moderation analyses.

In contrast to rumination and magnification, helplessness had no significant associations (direct or indirectly) with walking in relation to fatigue. This component of catastrophism has been especially studied in relation to pain. Specifically, previous studies have shown the relevance of helplessness as the best predictor of the affective dimension of pain [55] and disease impact in fibromyalgia patients with high pain [18] or high chronicity [53]. In other chronic pain populations, helplessness has been identified especially as a risk factor for functionality [56] as well as for persistence, severity and intensity of pain [25,54,57,58,59,60]. In agreement with previous literature, the results of the bivariate analyses showed that pain maintained the highest correlations with helplessness in comparison to the other components of catastrophism. Although, in a bivariate manner, helplessness also maintains significant correlations with fatigue, this association is diluted in the moderation analysis of walking, including disability as a covariable. Curiously, in this case, not even walking showed direct effects on fatigue (in comparison to what happened with the other two dimensions of catastrophism). Given that to the best of our knowledge there are no other studies in this regard, the current result should be the focus of further research. These preliminary results suggest that helplessness is not a variable of interest in the prediction of fatigue when walking is included and disability taken into account. Data also show the heterogeneity of the concept of catastrophism [24,55]. Therefore, although it would be necessary to carry out exhaustive analyses regarding the role of helplessness on fatigue, it seems as if the tendency is for helplessness to have a more important role in the prediction of pain and functionality than on fatigue.

As noted above, pain catastrophism has been widely and consistently associated with various symptoms in patients with fibromyalgia, including pain, depression, distress or cognitive and functional impairments [7,61,62,63,64]. Its relationship with fatigue, however, has been more rarely investigated, and the results to date are not consistent. For example, Thompson et al. [30] found significant associations between pain catastrophism and pain severity but not with disability or fatigue. In contrast, Lukkahatai et al. [28] revealed that fatigue was associated with several symptoms and psychological factors, including pain catastrophism. Our results suggest that the discrepancy in findings might be because catastrophism has been studied in a global and homogenous manner, and its different components have not been analyzed separately [24,25,54]. In fact, our findings show no significant results for catastrophism as a general construct but did find differential results when each dimension is studied separately. On the other hand, the discrepancies in previous literature could be due to the fact that the hypothesized linear association between catastrophizing components and fatigue might be more complex (i.e., contextually determined or moderated by a third factor) when a more contextual and functional perspective is taken into account [65]. This is consistent with past research [31] and with the psychological flexibility model of pain, the model which Acceptance and Commitment Therapy is based on, from the cognitive-evaluative perspective, in which two types of variables are considered to influence the outcome: (1) environmental variables that include direct experience, and (2) cognitive variables (verbal, language-based or cognitive processes) [65]. From this perspective, pain catastrophizing, as a cognitive variable would have a place within the aforementioned models [35,66].

Thus, the present study provides further support for this contextual relationship between psychological processes and outcomes in chronic pain. Our results have also shown a contribution of (adequate) adherence to walking to reduced fatigue, which is consistent with the findings of previous studies [11,12]. The novelty of the current investigation lies in the fact that the contribution of walking remained in multivariate models that included pain catastrophism (total scores and dimensions, except for helplessness) and in the fact that low adherence to walking imposed such a burden on patients who minimized the contribution of two of the three catastrophizing components, namely, rumination and magnification. Again, this supports the importance of walking and suggests that reducing rumination and magnification in patients with low adherence to walking is likely to be less effective for fatigue. Thus, in line with personalizing interventions, it is possible that if we want to maximize the effectiveness of an intervention to reduce rumination or magnification, adherence to walking should be encouraged first. This is speculative at this stage, but the fact that these two catastrophizing factors contributed to fatigue only when patients walked supports this idea. As noted earlier, according to our findings, this personalization is unlikely to be necessary when trying to reduce helplessness to minimize fatigue.

This study has some limitations. First, the associations must be interpreted according to the observational nature of the design, which does not allow causality inferences. Second, this study is based on self-reported data; in this sense, a relevant limitation is presented by the walking having been assessed using self-reports. It should be noted in this regard that the current study is part of a larger project that previously assessed this same behavior by means of self-administered questionnaires and pedometers, and a high consistency was found between measures [67,68,69]. An additional shortcoming is the homogeneity of the sample, which was composed of only fibromyalgia patients. Thus, there is no guarantee that the results can be generalized to other chronic pain populations due to the important differences between this population and other populations with chronic pain [37,63].

While this study has some shortcomings, the findings could have important clinical implications for practices in the field of psychological and interdisciplinary treatments and personalization in chronic pain [70] and in fibromyalgia [71]. The results support the benefits of walking as a form of physical exercise for fatigue, in this case after controlling for the influence of pain catastrophism. These results are consistent with the positive effects of walking on various fibromyalgia symptoms [8,9,10].

In relation to personalized treatments, our findings suggest that slightly different recommendations should be provided to reduce fatigue according to the walking pattern of individuals. In particular, among patients who walk, when attempting to regulate fatigue by reducing catastrophizing, tapping into the rumination and magnification components might be adequate according to our findings. In this regard, cognitive-behavioral therapy has proven to be effective in reducing these catastrophizing dimensions [72], and specific protocols for fibromyalgia patients are available [73]. In contrast, in the presence of patients who do not walk, it would probably be more advisable to promote adherence to walking by combining motivational and volitional strategies [14,67]. We encourage researchers to test the previous hypotheses in well-planned randomized controlled trials including fibromyalgia patients with low and high adherence to walking, including longitudinal designs.

## 5. Conclusions

As described in previous literature, walking is one of the most effective treatments in reducing the symptoms of patients with fibromyalgia. Nevertheless, the patients consistently show low adherence to walking, given that at the beginning patients frequently feel more pain and fatigue. The latter has scarcely been studied despite it being one of the more disturbing symptoms. Our results, in agreement with the cognitive flexibility models, suggest that interventions aimed at improving rumination and magnification could contribute to reducing fatigue among the patients who do walk and would therefore increase adherence to walking as a form of treatment and management of fatigue (as a factor inhibiting walking).

## Figures and Tables

**Table 1 ijerph-19-04198-t001:** Means, standard deviations and Pearson correlations between continuous study variables (fatigue, catastrophizing and its dimensions) and possible covariables (disability and pain).

	Mean (SD)	Magnification	Helplessness	Pain Catastrophism	Fatigue	Pain	Disability
Rumination	10.16 (4.04)	0.71 **	0.78 **	0.91 **	0.29 **	0.32 **	0.48 **
Magnification	6.61 (3.10)		0.74 **	0.87 **	0.18 **	0.26 **	0.43 **
Helplessness	15.01 (5.61)			0.95 **	0.35 **	0.38 **	0.54 **
Pain catastrophism	31.80 (11.68)				0.32 **	0.36 **	0.51 **
Fatigue	16.91 (2.90)					0.22 **	0.54 **
Pain	7.16 (1.52)						0.57 **
Disability	72.18 (16.97)						

** *p* < 0.01.

**Table 2 ijerph-19-04198-t002:** Mean differences between walking and non-walking groups in pain catastrophism, fatigue and covariables (disability and pain).

	Walking *n*= 132	Non-Walking *n* = 98	t	*p*	d-Cohen
Rumination	9.68 (4.18)	10.82 (3.75)	2.134	0.034	0.29
Magnification	6.29 (2.95)	7.04 (3.21)	1.810	0.072	0.24
Helplessness	14.04 (5.80)	16.44 (4.95)	3.269	0.001	0.45
Pain catastrophism	30.02 (11.92)	34.30 (10.73)	2.829	0.005	0.38
Fatigue	16.41 (3.15)	17.59 (2.40)	3.181	0.002	0.42
Pain	6.93 (1.54)	7.43 (1.45)	3.201	0.002	0.34
Disability	69.25 (17.5)	76.18 (15.44)	2.521	0.004	0.48

**Table 3 ijerph-19-04198-t003:** Mean differences between patients who took painkillers and who did not in pain catastrophism (and rumination, magnification and helplessness), fatigue and covariables (disability and pain).

	Painkillers				
	Yes *n* = 203	No *n* = 27	t	*p*	d-Cohen
Rumination	10.68 (3.77)	7.44 (4.51)	−4.092	0.000	0.78
Magnification	6.81 (3.08)	5.52 (2.81)	−2.216	0.033	0.43
Helplessness	15.41 (5.47)	12.81 (5.51)	−2.316	0.021	0.47
Pain catastrophism	32.9 (11.24)	25.77 (11.67)	−3.081	0.002	0.62
Fatigue	17.14 (2.62)	16.37 (2.63)	−1.445	0.150	-
Pain	7.29 (1.4)	6.36 (1.69)	−3139	0.002	0.59
Disability	73.97 (15.25)	62.91 (19.69)	−3407	0.001	0.62

**Table 4 ijerph-19-04198-t004:** Prospective prediction of fatigue from pain catastrophism, walking and their interaction.

	R2	F	*p*	Beta	t	*p*	95% Confidence Interval (CI)
DV = Fatigue	0.17	11.530	<0.001				
Pain Catastrophism				0.007	0.31	0.752	−0.05, 0.04
Walking				−1.86	−1.84	0.049	−3.85, 0.12
Interaction				0.04	1.52	0.129	<−0.01, 0.10
Disability				0.11	4.85	<0.001	0.07, 0.16
DV = Fatigue	0.15	10.062	<0.001				
Rumination				0.02	0.28	0.778	−0.15, 0.11
Walking				−2.06	−2.17	0.030	−3.92, −0.19
Interaction				0.16	1.96	0.049	<0.01, 0.33
Disability				0.14	4.23	<0.001	0.07, 0.19
DV = Fatigue	0.13	68.763	<0.001				
Magnification				0.08	1.05	0.294	−0.24, 0.07
Walking				−1.59	−2.06	0.041	−3.12, 0.01
Interaction				0.22	1.83	0.047	−0.03, 0.38
Disability				0.16	4.97	<0.001	0.09, 0.22
DV = Fatigue	0.15	10.009	<0.001				
Helplessness				0.07	1.40	0.161	<−0.03, 0.17
Walking				−0.59	−0.57	0.569	−2.61, 1.44
Interaction				0.02	0.35	0.725	−0.10, 0.14
Disability				0.12	3.79	<0.001	0.06, 0.19

**Table 5 ijerph-19-04198-t005:** Conditional effects of rumination/magnification on fatigue in function of walking (yes/no).

**Walking**	**Beta (Rumination)**	**t**	** *p* **	**95% CI**
No	−0.018	−0.282	0.778	−0.15, 0.11
Yes	0.146	2.671	0.008	0.04, 0.25
**Walking**	**Beta (Magnification)**	**t**	** *p* **	**95% CI**
No	0.032	0.481	0.630	−0.12, 0.16
Yes	0.216	4.005	<0.001	0.11, 0.32

## Data Availability

Not applicable.

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
