# Peer review of "The Role of Walking in the Relationship between Catastrophizing and Fatigue in Women with Fibromyalgia"

_ijerph, 2022, doi:10.3390/ijerph19074198_

Round 1

Reviewer 1 Report

Thank you for allowing me to review “The role of catastrophizing in the benefits of walking on fatigue in women with fibromyalgia.” The authors aim to describe the relationships between fatigue, walking behaviors, pain catastrophizing, and outcomes in patients with fibromyalgia. Although these variables are all important to examine, the theoretical underpinning of the study, as well as the data analyses, are fundamentally flawed, making the manuscript impossible to interpret as is. The cross-sectional methodology utilized is inappropriate for answering the central question of the manuscript.

Abstract

There is no theoretical link provided for the relationship between walking, fatigue, and pain catastrophizing. The authors just state they commonly occur together, without explaining how.

When the authors state that “Rumination and magnification contributed to more fatigue only when patients engaged in walking behavior” it sounds like walking is contraindicated. However, the conclusions contradict this finding.

Introduction

The authors provide no theoretical rational for associating fatigue with pain catastrophizing. It is critical that this be provided, given that it is the major focus of the paper.

The theoretical underpinning of the paper does not make sense, nor is it supported by literature. The authors assume that all of fatigue is tied to walking behavior (i.e., in the worst case scenario, when patients do not walk at all, fatigue will be so high that pain catastrophizing wont contribute to outcomes.) There is no support for this assertion. Also, even at very high levels of fatigue, pain catastrophizing may still make outcomes worse; especially considering possible correlations between fatigue and catastrophizing.

The authors do not describe the procedures at all. It is not clear what the participants did. As it stands, it appears that the participants just completed questionnaires. What else did they do at the visit? Was it conducted in person or online? What order were the questionnaires administered? How much were participants compensated? Was there a control group? What were the inclusion exclusion criteria and how were they determined? Who conducted the session, and where? How many assessment timepoints were there? If this is indeed a single assessment (i.e., cross-sectional data) then that is completely inappropriate for answering the primary question at hand, which is essentially a mediation question that needs 3 timepoints (effect of walking leads to fatigue via catastrophizing).

The authors are testing a moderation model when in reality they should be doing a mediation model (see above).

Results

The first paragraph of the results section should be deleted.

The authors to not control for disability, nor do they control for the high intercorrelations among study variables. The relationships may be explained by the fact that those with higher disability or pain intensity tend to walk less, and also have higher fatigue and catastrophizing. These covariates need to be included in analyses.

The authors never control for medication, despite the fact that rumination and fatigue can be common side effects of several medications.

Alternate models and potential third variables are not considered.

How were outliers handled? Were any outliers identified?

Reviewer 2 Report

Review comment

This manuscript entitled “The role of catastrophizing in the benefits of walking on fatigue in women with fibromyalgia” primarily aimed to explore the moderating effect of walking on the association between catastrophizing and fatigue. The topic will be of great interest to the readership. However, several questions should be addressed, which lists below.

Specific comments

  1. Abstract, ‘our goal is to explore the moderating effect of walking on the association between catastrophizing and fatigue’, change to ‘our goal was to…’. Authors should emphasize ‘fibromyalgia’ when presenting the purpose of the study.
  2. Keywords, please modify and improve the quality of the keywords as this will assist others when they are searching for information on your research topic. Avoid using ‘Fibromyalgia’, ‘walking’, ‘fatigue’, and ‘catastrophizing’ since they appear in the title.
  3. Introduction, ‘…when its diagnostic criteria were changed and the clinical significance of this comorbid symptomatology was clearly recognized.’, I suggest the authors provide more details about the diagnostic criteria and clinical significance of fibromyalgia here.
  4. ‘Pain catastrophizing is a highly prevalent coping strategy among fibromyalgia patients…’, Why is pain catastrophizing a coping strategy for people with fibromyalgia? But in the abstract, the authors explain it as a cognitive process. Please explain.
  5. Participants, whether the authors considered patient characteristic categories, such as regions of pain, in the subject inclusion process?
  6. Statistical analysis, please provide more details.
  7. Results, ‘This section may be divided by subheadings… as well as the experimental conclusions that can be drawn.’, Please remove this content.
  8. Discussion, ‘… it has been assumed that the relationship between pain catastrophizing and outcomes (i.e., fatigue) is linear (i.e., not con-textually determined).’, this sentence is a bit unclear, please consider modifying it.
  9. In summary, please make sure that your manuscript is properly prepared and formatted before submitting a revision.

Reviewer 3 Report

Dear authors,

thanks for your research.
The paper is well organized but some ameliorations are needed.
I suggest to introduce the concepts of magnification, helplessness and runmination in the introduction too.
The materials and methods are well presented.
Did you use SPSS as software for your statistical analysis?
The conclusions should be more clear.
The definition of helplessness (lines 203-204) should be moved to the introduction. 

Regards
